# Development of Semisynthetic Apoptosis-Inducing Agents Based on Natural Phenolic Acids Scaffold: Design, Synthesis and In-Vitro Biological Evaluation

**DOI:** 10.3390/molecules27196724

**Published:** 2022-10-09

**Authors:** Shahira M. Ezzat, Heba El Sayed Teba, Inas G. Shahin, Ahmed M. Hafez, Aliaa M. Kamal, Nora M. Aborehab

**Affiliations:** 1Pharmacognosy Department, Faculty of Pharmacy, Cairo University, Cairo 11562, Egypt; 2Pharmacognosy Department, Faculty of Pharmacy, October University for Modern Sciences and Arts (MSA), Giza 12451, Egypt; 3Organic Chemistry Department, Faculty of Pharmacy, October University for Modern Sciences and Arts (MSA), Giza 12451, Egypt; 4Department of Biochemistry, School of Life and Medical Sciences, University of Hertfordshire Hosted by Global Academic Foundation, Cairo 11586, Egypt; 5Pharmaceutical Organic Chemistry Department, Faculty of Pharmacy, Cairo University, Cairo 11562, Egypt; 6Biochemistry Department, Faculty of Pharmacy, October University for Modern Sciences and Arts (MSA), Giza 12451, Egypt

**Keywords:** anti-proliferative activity, synthesis, anticancer activity, cancer cell lines, drug discovery, semisynthetic amides, apoptosis inducer agents, Caco-2 cell line, matrix metalloproteinases

## Abstract

A crucial target in drug research is magnifying efficacy and decreasing toxicity. Therefore, using natural active constituents as precursors will enhance both safety and biological activities. Despite having many pharmacological activities, caffeic and ferulic acids showed limited clinical usage due to their poor bioavailability and fast elimination. Therefore, semisynthetic compounds from these two acids were prepared and screened as anticancer agents. In this study, CA and FA showed very potent anticancer activity against Caco-2 cells. Consequently, eighteen derivatives were tested against the same cell line. Four potent candidates were selected for determination of the selectivity index, where compound **10** revealed a high safety margin. Compound **10** represented a new scaffold and showed significant cytotoxic activity against Caco-2. Cell-cycle analysis and evaluation of apoptosis showed that derivatives **10**, **7**, **11**, **15** and **14** showed the highest proportion of cells in a late apoptotic stage.

## 1. Introduction

Cancer has the highest social and economic burden among all human diseases, and coupled with that, in the coming four decades it is anticipated to leap from the second to the first place to become the leading cause of death worldwide [1].

Over the years, several natural anticancer molecules were used as scaffolds for the synthesis of anticancer drugs, such as vinblastine and vincristine from *Catharanthus* alkaloids [2], paclitaxel from Yew trees [3], and podophyllotoxin from Podophyllum resins [4].

Natural phenolic compounds are an important class of substances of plant origin that have both anticancer and anti-oxidant activities [5,6,7,8]. For this reason, new methods of their functionalization are developed to increase their application in medicine. Two of the most prevailing phenolic compounds are caffeic acid (CA) and ferulic acid (FA) (Figure 1).

Caffeic acid and its different derivatives were reported to possess many biological activities such as immunostimulants [9,10,11,12], antibacterial [13,14], anti-inflammatory [15], cardio protective [15,16], anti-atherosclerotic [9,17], antidiabetic [15,18], antiproliferative [19,20], hepatoprotective [21], anticancer [9,13,14,15,22,23,24], and Alzheimer’s disease [25].

Ferulic acid has been used in healthy food and nutrition restoratives due to its well-known antioxidant properties, and due to this biological effect, it can be used in Alzheimer’s disease [25,26]. It is highly recommended for diabetic patients [27]. Moreover, FA has a reported neuroprotective effect [28,29,30] and relieves ischemia-reperfusion-induced cell apoptosis [31,32]. FA also has a chemo–preventive activity against carcinogenesis induced by 7,12-dimethylbenz[*a*]anthracene (DMBA) in rats [33,34]. Equally important is its antiproliferative action on osteosarcoma cells where it induces apoptosis by blocking the PI3K/Akt pathway [35].

It is highly recommended for diabetic patients as it can alleviate oxidative stress and also has an antihyperglycemic effect. [27]

Recently, several phenolic compounds were tested against a spectrum of severe pathological conditions and both CA and FA showed potential results against coronavirus-based infections [36].

The aforementioned biological activities are achieved through a variety of mechanisms; the anticancer activity of both CA and FA is exerted by two mechanisms, acting as primary antioxidants that eliminate reactive oxygen species (ROS) and as secondary antioxidants (pro-oxidants) preventing the splitting of peroxides by metal ion chelation (iron and copper) [37,38]. Thus, inhibiting oxidative degradation of essential biological molecules (lipids, amino acids, polyunsaturated fatty acids and DNA bases) and preventing the formation of cellular lesions [25,39,40,41,42].

As antioxidants, the study of the structure activity relationship of both acids and their derivatives showed that the hydroxy groups in caffeic and ferulic acids proved to be a pre-requisite for anti-oxidant activity, through their free radical scavenging and metal chelation properties. The 4-hydroxy moiety supports the resonance around the whole molecule resulting in higher stability due to the formation of the phenoxy radical. It is also accepted that a second 3-hydroxy substitution in caffeic acid promotes the antioxidant quality due to the formation of a stable ortho-quinone structure. Equally, in ferulic acid, the methoxy group enhances the resonance of the aromatic ring due to its electron-donating attributes. The acids have an additional aspect where the COOH tail provides lipid support and resonates with its neighboring unsaturation, offering alternative sites to prevent membrane deterioration by free radicals while the inclusion of H-donating groups in the side chain such as amide moiety accelerates the activity [43,44,45] (Figure 2).

As cell apoptosis promoters, it was established that CA and FA play a distinctive role in encouraging cell apoptosis in human cervical cancer cells through inhibition of the Bcl-2 activity [46,47] and in non-small lung cancer cells (NSCLC) through the MAPK pathway [48]

In another pathway, it was observed that the acids are responsible for suppressing hepatocellular carcinoma cells (HCC) via inhibiting the expression of vascular endothelial growth factor (VEGF), which in turn impedes angiogenesis of the tumor cells [49].

It is worth mentioning that both CA and FA are highly selective, imparting cytotoxicity to tumor cells but not to normal cells; on the other hand, they are highly metabolized in the body, therefore jeopardizing their stability. By evaluating the two major derivatives—amides and esters—the amides possessed potent cytotoxic activity in addition to high in-vivo stability when put in comparison with the ester derivatives. Following in the same vein, 18 amide compounds (Table 1) were synthesized (Figure 3) via reacting the respective phenolic acids with a panel of amines, adopting N,N’-dicyclohexylcarbodiimide (DCC) as a coupling agent.

Therefore, the current study was conducted to verify the anti-tumor activity of CA and FA versus their semisynthetic derivatives on different cell lines and explore the underlying mechanisms using the following scheme:

## 2. Results and Discussion

### 2.1. Chemistry

The synthesis of CA and FA derivatives **3**–**20** is displayed in Figure 3. Preparation of these candidates was carried out by reacting the naturally obtained caffeic or ferulic acids with different amines using N,N’-dicyclohexylcarbodiimide (DCC) as a coupling agent for the preparation of the amide linkage starting from the acids. Separated amides **3**–**20** were purified by column chromatography. The main idea is screening the effect of changing the alkyl/cycloalkyl/aralkyl/aryl/heterocyclic moiety of the amide functionality on the biological activity. Compounds **1** and **2** exhibited a sky-blue color in UV, which turns to a greenish yellow color on exposure to ammonia vapor and spraying with AlCl_3,_ indicating that they are phenolic acid in nature. ^1^H NMR spectrum of **1** showed the characteristic signals for caffeic acid [50], which was characterized by two trans-olefinic protons at δ 6.16 and δ 7.41 ppm with a large coupling constant δ 15.88 Hz, assigned to H-8 and H-7, respectively. An ABX system was displayed at δ 6.76, 6.95 and δ 7.04 ppm assigned to H-5, H-6 and H-2, respectively. The ^13^C NMR showed the nine characteristic carbon signals for 3,4-dihydroxy trans cinnamic acid with two oxy carbons at δ 146.0 and δ 148.5 (C-3 and 4), one carbonyl carbon at δ 168.3 (C-9), two trans olefinic carbons at δ 145.07 and δ 115.5 ppm, in addition to C-1 at δ 126.1, C-2 at δ 115.1, C-5 at δ 116.2 and C-6 at δ 21.6. ^1^H NMR spectrum of **2** showed the characteristic signals for ferulic acid [51], which was characterized by two trans-olefinic protons at δ 6.35 and 7.48 ppm with large coupling constant 15.88 Hz, assigned to H-8 and H-7, respectively. An ABX system was affirmed by these signals at δ 6.79, 7.07 and 7.28 ppm, corresponding to H-5, H-6 and H-2, respectively. The methoxy group at C-3 was manifested by the singlet signal at δ 3.82 ppm and integrated as three protons (Appendix A).

The ^13^C-NMR supported the nine characteristic carbon signals for 4-Hydroxy-3-methoxycinnamic acid with two oxy carbons at δ 148.3 and 149.5 ppm (C-3 and C- 4), one carbonyl carbon at 168.4 ppm (C-9), two trans olefinic carbons at δ 144.9 and δ 115.9 ppm and finally, C-1, C-2, C-5 and C-6 were confirmed by signals at δ 126.2, 111.6, 116.0 and 123.2 ppm respectively.

Reacting **1** and **2** with either phenyl piperazine, cyclohexyl amine, aromatic amines or aralkyl amine in presence of DCC afforded compounds **3**, **12**, **6**, **15**, **4**, **5**, **7**, **8**, **10**, **11**, **13**, **14**, **16**, **17**, **19**, **20**, **9** and **18** sequentially. The structures of these compounds were elucidated using spectral data and microanalyses.

IR spectra of all the candidates showed the characteristic carbonyl band of amide between 1626–1662 cm^−1^ while the NH band appeared in the range of 3510–3240 cm^−1^ for all compounds except **3** and **12,** which are tertiary amides. ^1^H NMR spectra of compounds **3**–**20** exhibited the appearance of the doublet signal of HC=CH in the range of δ 5.70–7.73 ppm with large coupling constant 15.00–15.90 Hz. Compounds **3** and **12** revealed the multiplet signals in their ^1^H NMR spectra at δ 1.04–1.85 ppm assigned to CH_2_ of piperazine ring. ^1^H NMR spectra of compounds **5**, **14** and **17** showed a series of singlet signals in the range of δ 2.26–2.29 ppm due to CH_3_ group while the 11 protons corresponding to the cyclohexyl group in compounds **6** and **15** were confirmed by the appearance of multiplet signals in the range of δ 1–3.35 ppm. Compound **8** was the only CA derivative that showed a singlet signal corresponding to OCH_3_ group at δ 3.63 ppm while **16** showed an additional OCH_3_ group at δ 3.81 ppm. The appearance of singlet signals at δ 5.59 and 3.19 ppm in ^1^H NMR spectra of **9** and **18** respectively confirmed the presence of N-CH_2_ (Appendix A). Compounds **11** and **20** revealed the presence of NH_2_ group at δ 9.17 and 4.94 ppm, respectively. ^13^C NMR spectrum of all compounds confirmed the presence of the characteristic C=O group, showing signals in the range of δ 160.7–170.9 ppm.

### 2.2. Biological Activity

#### 2.2.1. Cell Viability Determination by MTT Assay

The cytotoxicity of the parent compounds **1** and **2** was tested against breast (MCF-semi-synthesized compounds on Caco-2 cancer cell line and the 7), liver (HepG2), lung epithelial (A459), colorectal (Caco-2) and pancreatic (PANC-1) cancer cell lines. Their cytotoxicity was compared to doxorubicin (DOX) and their selectivity was tested on Vero cells (Table 2). The cytotoxicity of both **1** and **2** was lower than that of DOX but the selectivity index (SI) was significantly higher than DOX (Appendix A).

Caco-2 showed the highest sensitivity to both acids **1** and **2** (IC_50_ = 0.41 μM for both caffeic and ferulic acids) and thus it was selected for testing the cytotoxicity of the 18 semisynthetic candidates. All the synthesized amides were far more potent than the parent acids by 10-fold or more with a slight reduction in the SI of these derivatives (Table 3). It is worth mentioning that caffeic acid derivatives were more potent than ferulic acid derivatives. Among the CA amide derivatives **3**–**11**, compound **10** was the most potent against the Caco-2 cancer cell line with an IC_50_ of 0.009 ± 0.09 μM with a SI 14-fold greater than its IC_50_ followed by compounds **11**, **7** and **9** with an IC_50_ of 0.01 ± 0.11, 0.02 ± 0.11 and 0.03 ± 0.29 μM, respectively. The potency of the rest of the CA amide derivatives ranged from IC_50_ = 0.05 μM to 0.08 μM, which was still much more potent than the parent acid. Regarding the FA amide derivatives **12**–**20**, compound **20** revealed the highest potency with an IC_50_ of 0.01 ± 0.63 μM followed by compounds **15**, **17** and **19** with IC_50_ of 0.02 ± 0.17, 0.02 ± 0.21 and 0.02 ± 0.65 μM sequentially. Moreover, compounds **12**, **13**, **14**, **16** and **18** showed very good cytotoxicity against Caco-2 with an IC_50_ ranging from 0.04 μM to 0.08 μM. The derivatives were much more potent than the parent acid with an excellent SI (Table 3). To further confirm this, cell-cycle analysis and apoptosis assays were performed and the gene expression of p53, caspase 3, Bax, MMP-2 and MMP-9 was quantified.

#### 2.2.2. Cell Cycle Progression

The potential candidates with anti-tumor effect on the Caco-2 cell line along with a high SI were chosen for cell-cycle analysis. The cell-cycle distribution and cell death were analyzed using Annexin V-FITC/PI staining and flow cytometry (Table 4, Figure 4). The potential candidates that showed potent anticancer activity against the Caco-2 cell line and high SI were chosen to investigate how they will change the cell-cycle distribution.

##### Flow Cytometry Analysis Showing Induction of Apoptosis

Cell-cycle analysis showed that the treatment of Caco-2 cells with the parent compounds and their derivatives resulted in an increase in the cell population in the G2/M phase, indicating apoptosis. In the case of caffeic acid derivatives, the highest induction of apoptosis was seen in compounds **10**, **7** and **11** (23.42%, 18.26% and 14.04%), respectively, which showed a more significant increase than the caffeic acid itself (11.31%). On the contrary, compounds **5** and **9** showed a non-significant increase in the apoptotic population compared to the parent compound **1** (Table 4).

Concerning Ferulic acid derivatives, the accumulation in the Pre-G1 phase and thus the highest induction of apoptosis was observed with candidates **15** and **14** (25.17% and 16.51% respectively) which showed a more significant effect than the parent compound **2** itself (13.22%), whereas the derivatives **17**, **18** and **20** showed non-significant decrease in the apoptotic population compared to parent compound **2.** (Table 4)

##### Evaluation of Apoptosis by Annexin V-FITC Staining

Quantification of the different types of apoptotic cells was of our interest due to the accumulation of cells at the G2/M phase during cell-cycle analysis following the treatment with the parent compounds **1** and **2** and their potential candidates.

After Caco-2 cells were treated with parent compounds **1** and **2** and their derivatives, they were stained with annexin V-FITC/PI, and then flow cytometry was used to detect the cell-cycle distribution as shown in Figure 5 and Table 5.

The results of the evaluation of apoptosis revealed that potential candidates **7**, **10**, **11**, **14** and **15** showed the highest proportion of cells in the late apoptotic stage (stained by Annexin V-FITC and PI).

#### 2.2.3. Gene Expression

##### Caffeic Acid and Ferulic Acid Derivatives Enhanced Pro-Apoptotic Genes

The effect of caffeic acid **1** and its amide derivatives **5**, **7**, **9**, **10** and **11** as well as ferulic acid **2** and its derivatives **14**, **15**, **17**, **18**, and **20** on the expression of pro-apoptotic genes were tested. β-actin gene was used as a control gene. Gene expression was chosen over Western blotting for more accurate quantification and thus comparison. The gene expression of the p53, caspase 3 and Bax genes was measured. Most of the derivatives showed an increased expression of the three genes, which was evidenced by increasing apoptotic effects. In the case of caffeic acid derivatives, compound **10** showed the highest overexpression with an increase of 14- to 16-fold in the three genes, followed by **7** and **11** (Figure 6). Regarding the ferulic acid derivatives, **15** showed the highest overexpression of 10-fold in the p53 and 15-fold in the caspase 3 and Bax genes compared to the parent acid (Figure 7).

##### Caffeic Acid and Ferulic Acid Derivatives Inhibit Metastasis Enhancing Genes

Matrix metalloproteinases (MMPs) appear to play an important role in the formation and progression of human cancers. MMPs are involved in the breakdown of the extracellular matrix, which is a crucial stage in tumor invasion and spread [52]. A literature review revealed that both **1** and **2** decrease the expression of both MMP-2 and 9, which may lead to decrease in enzymatic activity [53].

To assess the effect of **1**, **2** and their derivatives **5**, **7**, **9**, **10**, **11**, **14**, **15**, **17**, **18**, and **20** on tumor progression and metastasis, the expression of the MMP2 and MMP9 was measured [54,55].

The derivatives of both **1** and **2** showed a significant decrease in the expression of both MMP2 and MMP9. Out of the most potential candidates, **17** and **7** showed the greatest inhibition in both groups (Figure 8 and Figure 9).

### 2.3. Structure–Activity Relationship Correlation

The parent phenolic acids **1** and **2** were tested against five cancer cell lines: Caco-2, A549, PANC-1, MCF7, and Hep G2, as well as one normal Vero cell line using DOX as positive reference. The parent compounds showed significant antitumor activity against Caco-2 cell line and with excellent selectivity index (SI). Consequently, all the newly synthesized amide derivatives **3**–**20** were evaluated for their cytotoxic activity against Caco-2 cancer cell line. The SI of amide derivatives was determined using the Vero normal cell line. They showed excellent cytotoxic activity, much more potent than the parent acids **1** and **2**. Based on the phenolic acids scaffold, the amide derivatives were designed to have all the features of both acids as well as the stable amide linkage. All the amide derivatives **3**–**20** showed very promising anticancer activity against Caco-2 cell line and very good SI. So, changing the carboxylic acid moiety into an amide functionality increased the biological activity and did not destroy the selectivity index of the parent compounds. Examining the structure of the six most active compounds (**7**, **10**, **11**, **14**, **15** and **17**) (Figure 10), starting from the anticancer activity against Caco-2 cell line and excellent SI, followed by cell-cycle analysis and apoptosis assays, then the effect of these candidates on gene expression of p53, caspase 3, Bax, MMP-2 and MMP-9, it was found that besides all the features of both acids and the amide linkage, five compounds out of six have aromatic amide moieties. The presence of *p*-chloro on the phenyl ring of the amide linkage resulted in the most cytotoxic compound of all the six candidates. Replacing the chloro atom in compound **10** with bromo in compound **7** decreases the antitumor activity. Moreover, it was found that the presence of the cycloalkyl group of the amide linkage in the case of compound **15** increased the induction of apoptosis and showed the highest proportion of cells in late apoptotic stage. In case of enhancing the pro apoptotic genes (p53, caspase 3 and Bax), compound **10** (with *p*-chlorophenyl) revealed the highest activity. Furthermore, in case of inhibiting the metastasis-enhancing genes, compound **17** with *m*-tolyl group was the most active one.

## 3. Materials and Methods

### 3.1. General

Silica gel 60 (70–230 mesh ASTM; Fluka, Steinheim, Germany), Diaion HP-20 AG, Sephadex LH-20 (Pharmacia Fine Chemicals AB, Uppsala, Sweden) and silica gel GF_254_ precoated thin layer plates (Fluka, Steinheim, Germany) were used. Chromatograms detections were performed under UV light (at 254 and 366 nm) and sprayed by *p*-anisaldehyde sulphuric acid spray reagent. Melting points (°C, uncorrected) were determined on Stuart apparatus and the given values were uncorrected. Progress of the reaction was monitored by thin layer chromatography (TLC) using aluminum sheets precoated with UV fluorescent silica gel (MERCK, Rahway, NJ, USA, 60 F 254), and spots were visualized by UV lamp. The solvent system used was dichloromethane: methanol (in different ratios). The IR spectra (cm^−1^) were determined using KBr discs on a Shimadzu IR8400s Spectrophotometer (Microanalytical Unit, Faculty of Pharmacy, Cairo University, Cairo, Egypt). ^1^H NMR and ^13^C NMR spectra were performed on Bruker 400-BB 400 MHz Spectrophotometer, microanalytical unit, Faculty of Pharmacy, Cairo University, Cairo, Egypt, using tetramethylsilane (TMS) as internal standard, and chemical shift values were recorded in ppm on *δ* scales. Peak multiplicities were designed as follows: s, singlet; d, doublet; t, triplet; m, multiplet. Elemental analyses were performed at the Regional Center for Mycology and Biotechnology, Faculty of Pharmacy, Al-Azhar University, Cairo, Egypt. The data were statistically evaluated using the Graph Pad Prism 6 program. Data were reported as mean ± SD of the triplicates of each experiment. One-way ANOVA with multiple comparisons post-hoc tests were used for analysis. Results were considered significant at *p* < 0.05.

### 3.2. Plant Material

The grains of oats (*Avena sativa* L. Family Poaceae) were obtained from the Agricultural Research Center, Giza, Egypt. The plants were authenticated at Botany Department, Faculty of Science, Cairo University, Giza, Egypt. Voucher samples no (3-10-2018) of the plants were deposited at the Museum of the Pharmacognosy Department, Faculty of Pharmacy, Cairo University.

#### Extraction and Isolation of Caffeic and Ferulic Acids

The powdered oats (2 kg) were extracted with 80% ethanol at 55 °C for 165 min (14 L × three times), the ethanolic extract was evaporated under vacuum at 40 °C. The ethanolic residue (350 gm) was suspended in distilled water (1 L) and defatted by methylene chloride (1 L × 3 times), then the defatted aqueous solution was fractionated over a Dianion column using 100% water, methanol-water (1:1 *v*/*v*) and 100% methanol. The solvent in each case was evaporated under vacuum to obtain three fractions 100% water fraction (75 gm), MeOH-water (1:1) fraction (55 gm) and 100% methanol fraction 67 gm.

Part of the 100% methanol fraction (5 g) was purified was chromatographed over a sephadex LH-20 column (3 × 30 cm), eluted with gradient elution 50%, 80% and 100% methanol (300 mL each). Fractions (10 mL, each) were collected and monitored by TLC (silica gel GF_254_ precoated plates- Fluka) using solvent system (dichloromethane-methanol 9:1 *v*/*v*). Similar fractions were pooled together to obtain two pure compounds **1** (Rf 0.48, 500 mg, white crystals) and **2** (Rf 0.55, 750 mg, white crystals). This process was repeated several times to obtain 10 g of each compound.

### 3.3. Chemistry:

#### 3.3.1. General method of Preparation of Caffeic or Ferulic Acid Amide Derivatives (3–20)

A mixture of either caffeic acid **1** or ferulic acid **2** (0.005 mol), the appropriate amine (0.005 mol) and DCC (1.03 g, 0.005 mol) in anhydrous THF (5 mL) was stirred in an ice-bath for an hour then left to stir at room temperature for 24 h. The mixture was filtered off then the filtrate was left to evaporate to dryness to provide compounds **3**–**20**. Purification of the final compounds was carried out using either crystallization or a glass column (2 × 30 cm) of a sephadex LH-20 (Pharmacia Fine Chemicals AB, Uppsala, Sweden), through using (100% methanol) as eluent. Fractions 1 mL were collected and measured through using the TLC (silica gel GF254 precoated plates-Fluka) using solvent system (n-hexane-ethyl acetate 8:2 *v*/*v*).

##### (E)-3-(3,4-Dihydroxyphenyl)-1-(4-phenylpiperazin-1-yl)prop-2-en-1-one (3)

IR (KBr, cm^−1^): 3475 (OH), 3186 (C-H aromatic), 2944 (C-H aliphatic), 1639 (C=O). ^1^H NMR (DMSO-d_6_, D_2_O, δ): 1.36–1.85 (m, 8H, 2 N-CH_2_-CH_2_), 6.76–7.26 [m, 10H (2OH, D_2_O exchangeable + 8H Ar-H)], 7.35 and 7.39 (d, 2H, *J_value_* = 15.1MHz, HC=CH); ^13^C NMR (100 MHz, DMSO-d_6_, δ): 168.7, 149.8, 148.8, 141.8, 128.4, 127.6, 126.9, 125.9, 124.7, 116.9, 53.8, 49.6. Anal. Calcd. For C_19_H_20_N_2_O_3_ (324.37): C, 70.35; H, 6.21; N, 8.64. Found: C, 70.30; H, 6.30; N, 8.34.

##### (E)-3-(3,4-Dihydroxyphenyl)-N-(4-hydroxyphenyl)acrylamide (4)

IR (KBr, cm^−1^): 3460, 3355, 3323, 3298 (OH and NH), 3130 (C-H aromatic), 1646 (C=O). ^1^H NMR (DMSO-d_6_, D_2_O, δ): 6.40–6.49 (m, 4H, Ar), 6.59 and 6.84 (d, 2H, *J_value_* = 15 MHz, CH=CH), 7.42 (s, 1H, NH), 7.46–7.50 (m, 3H, Ar), 9.86 (s,3H, 3OH). ^13^C NMR (100 MHz, DMSO-d_6_, δ): 170.2, 149.8, 147.9, 143.3, 138.5, 136.1, 127.1, 126.1, 129.1, 114.2, 112.6. Anal. Calcd. For C_15_H_13_NO_4_ (271.26): C, 66.41; H, 4.83; N, 5.16. Found: C, 66.11; H, 4.53; N, 5.30.

##### (E)-3-(3,4-Dihydroxyphenyl)-N-(p-tolyl)acrylamide (5)

IR (KBr, cm^−1^): 3510, 3420, 3348 (OH and NH), 3140 (C-H aromatic), 2910 (C-H aliphatic), 1652 (C=O). ^1^H NMR (DMSO-d_6_, D_2_O, δ): 2.26 (s, 3H, CH_3_), 6.66–6.46 (m, 4H, Ar), 6.81 and 7.13 (d, 2H, *J_value_* = 14.9 MHz, CH=CH), 7.50 (s, 1H, NH), 7.60–7.43 (m, 3H, Ar), 10.0 (s, 2H, 2OH). ^13^C NMR (100 MHz, DMSO-d_6_, δ): 165.5, 149.5, 147.2, 143.3, 138.1, 137.5, 129.4, 128.6, 127.1, 125.3, 124.7, 114.7, 111.9, 21.7. Anal. Calcd. For C_16_H_15_NO_3_ (269.29): C, 71.36; H, 5.61; N, 5.20. Found: C, 71.66; H, 5.81; N, 5.44.

##### (E)-N-Cyclohexyl-3-(3,4-dihydroxyphenyl)acrylamide (6)

IR (KBr, cm^−1^): 3467, 3345, 3360 (OH and NH), 2956 (C-H aliphatic), 1667 (C=O). ^1^H NMR (DMSO-d_6_, D_2_O, δ): 1.00–1.76 (m, 10H, CH_2_), 3.42 (m, 1H, CH), 5.70 and 5.72 (d, 2H, *J_value_* = 15 MHz HC=CH), 6.76–7.26 (m, 3H, Ar-H + 2OH, NH, D_2_O exchangeable). ^13^C NMR (100 MHz, DMSO-d_6_, δ): 166.8, 148.9, 147.5, 143.9, 138.9, 129.4, 127.3, 125.1, 123.3, 122.7, 116.7, 110.9, 51.7, 31.5, 25.9; Anal. Calcd. For C_15_H_19_NO_3_ (261.31): C, 68.94; H, 7.33; N, 5.36. Found: C, 68.74; H, 7.53; N, 5.16.

##### (E)-N-(4-Bromophenyl)-3-(3,4-dihydroxyphenyl)acrylamide (7)

IR (KBr, cm^−1^): 3413, 3327, 3317 (OH and NH), 3200 (C-H aromatic), 1662 (C=O). ^1^H NMR (DMSO-d_6_, D_2_O, δ): 6.50–7.51 (m, 9H, 2OH D_2_O exchangeable + Ar-H), 7.58 and 7.66 (d, 2H, *J_value_* = 15.0 MHz HC=CH), 10.17 (s, 1H, NH) ppm. ^13^C NMR (100 MHz, DMSO-d_6_, δ): 170.9, 148.9, 146.3, 145.3, 136.3, 131.5, 128.9, 127.9, 126.8, 126.1, 121.7, 115.1, 111.9. Anal. Calcd. For C_15_H_12_BrNO_3_, (334.16): C, 53.91; H, 3.62; Br, 23.91; N, 4.19. Found: C, 53.71; H, 3.69; N, 4.30.

##### (E)-N-(4-Methoxyphenyl)-3-(3,4-dihydroxyphenyl)acrylamide (8)

IR (KBr, cm^−1^): 3510, 3468, 3420 (OH and NH), 3130 (C-H aromatic), 1646 (C=O). ^1^H NMR (DMSO-d_6_, D_2_O, δ): 3.63 (s, 3H, OCH_3_), 6.50–7.40 (m, 7H, Ar-H), 7.60 and 7.62 (d, 2H, *J_value_* = 14.8 MHz, HC=CH), 9.1–9.44 (s, 2H, 2OH, D_2_O exchangeable), 9.94 (s, 1H, NH D_2_O exchangeable) ppm. ^13^C NMR (100 MHz, DMSO-d_6_, δ): 165.9, 148.9, 146.3, 143.3, 139.3, 135.5, 129.9, 127.9, 126.8, 125.1, 121.7, 114.1, 113.9, 56.9. Anal. Calcd. For C_16_H_15_NO_4_ (285.29): C, 67.36; H,5.30; N, 4.91. Found: C, 67.56; H,5.39; N, 4.71.

##### (E)-N-Benzyl-3-(3,4-dihydroxyphenyl)acrylamide (9)

IR (KBr, cm^−1^): 3300–3350 (OH and NH), 3135 (C-H aromatic), 2927 (C-H aliphatic), 1626 (C=O). ^1^H NMR (DMSO-d_6_, D_2_O, δ): 5.59 (CH_2_), 6.52 and 6.56 (d, 2H, *J_value_* = 15.3 MHz, HC=CH), 6.77–7.51 (m, 8H, Ar-H), 9.21 and 9.45 (2H, OH, D_2_O exchangeable), 9.98 (s, 1H, NH, D_2_O exchangeable), ^13^C NMR (100 MHz, DMSO-d_6_, δ): 167.9, 146.9, 145.3, 141.3, 137.3, 128.9, 127.2, 126.8, 126.4, 123.7, 115.1, 112.9, 43.9. Anal. Calcd. For C_16_H_15_NO_3_ (269.29): C, 71.36; H, 5.61; N, 5.20. Found: C, 79.36; H, 5.81; N, 5.34.

##### (E)-N-(4-Chlorophenyl)-3-(3,4-dihydroxyphenyl)acrylamide (10)

IR (KBr, cm^−1^): 3510, 3413, 3317 (OH and NH), 3200 (C-H aromatic), 1635 (C=O). ^1^H NMR (DMSO-d_6_, D_2_O, δ): 6.50–7.02 [m, 9H (2OH, D_2_O exchangeable + 7H, Ar-H)], 7.71 and 7.73 (d, 2H, *J_value_* = 14.8 MHz, HC=CH), 10.209 (s, 1H, NH) ppm, ^13^C NMR (100 MHz, DMSO-d_6_, δ): 169.9, 148.5, 146.9, 144.3, 143.3, 133.3, 128.9, 127.2, 126.1, 124.7, 123.7, 117.9, 117.1, 113.9. Anal. Calcd. For C_15_H_12_ClNO_3_ (289.71): C, 62.19; H, 4.17; N, 4.83. Found: C, 62.28; H, 4.30; N, 4.89.

##### (E)-N-(2-Aminophenyl)-3-(3,4-dihydroxyphenyl)acrylamide (11)

IR (KBr, cm^−1^): 3500–3240 (OH, NH and NH_2_), 3200 (C-H aromatic), 1662 (C=O). ^1^H NMR (DMSO-d_6_, D_2_O, δ): 5.59 (CH_2_), 6.52 and 6.56 (d, 2H, *J_value_* = 15.3 MHz, HC=CH), 6.27–7.52 (m, 8 H, Ar-H), 9.17 (s, 2H, NH_2_, D_2_O exchangeable), 9.27 (s, 1H, NH, D_2_O exchangeable), 9.43 and 9.48 (2H, OH, D_2_O exchangeable). ^13^C NMR (100 MHz, DMSO-d_6_, δ): 168.7, 149.5, 146.9, 145.3, 141.3, 128.4, 127.6, 125.3, 124.5, 122.7, 117.2, 118.9, 114.9. Anal. Calcd. For C_15_H_14_N_2_O_3_ (270.28): C, 66.66; H, 5.22; N, 10.36. Found: C, 66.37; H, 5.02; N, 10.44.

##### (E)-3-(3-Hydroxy-4-methoxyphenyl)-1-(4-phenylpiperazin-1-yl)prop-2-en-1-one (12)

IR (KBr, cm^−1^): 3435 (OH), 3140 (C-H aromatic), 2 910 (C-H aliphatic), 1629 (C=O). ^1^H NMR (DMSO-d_6_, D_2_O, δ): 1.04–1.81 (m, 8H, 2 N-CH_2_-CH_2_), 3.36 (s, 3H, OCH_3_), 6.78–6.74 (m, 3H, Ar), 6.90 and 6.98 (d, 2H, *J_value_* = 15.2 MHz, CH=CH), 7.24–7.18 (m, 5H, Ar), 8.08 (s, 1H, OH). ^13^C NMR (100 MHz, DMSO-d_6_, δ): 160.7, 149.5, 148.9, 141.3, 128.4, 127.6, 126.1, 125.3, 124.7, 118.9, 55.7, 53.1, 48.6. Anal. Calcd. For C_20_H_22_N_2_O_3_ (338.40): C,70.99; H, 6.55; N, 8.28. Found: C,70.73; H, 6.61; N, 8.31.

##### (E)-3-(3-Hydroxy-4-methoxyphenyl)-N-(4-methoxyphenyl)acrylamide (13)

IR (KBr, cm^−1^): 3379, 3116 (OH and NH), 3066 (C-H aromatic), 2924 (C-H aliphatic), 1651 (C=O), ^1^H NMR (DMSO-d_6_, D_2_O, δ): 3.83 (s, 3H, OCH_3_), 6.40–6.49 (m, 4H, Ar), 6.59 and 6.84 (d, 2H, *J_value_* = 15.1 MHz, CH=CH), 7.42 (s, 1H, NH), 7.46–7.50 (m, 3H, Ar), 9.86 (s, 2H, OH). ^13^C NMR (100 MHz, DMSO-d_6_, δ): 169.7, 149.5, 148.9, 142.3, 138.1, 136.5, 126.1, 125.3, 129.7, 114.5, 112.9, 55.7, 53.7. Anal. Calcd. For C_16_H_15_NO_4_ (285.29): C, 67.36; H, 5.30; N, 4.91. Found: C, 67.21; H, 5.43; N, 4.82.

##### (E)-3-(3-Hydroxy-4-methoxyphenyl)-N-(p-tolyl)acrylamide (14)

IR (KBr, cm^−1^): 3460, 3290 (OH and NH), 3089 (C-H aromatic), 2846 (C-H aliphatic), 1645 (C=O). ^1^H NMR (DMSO-d_6_, D_2_O, δ): 2.26 (s, 3H, CH_3_), 3.83 (s, 3H, OCH_3_), 6.46–6.66 (m, 4H, Ar), 6.81 and 7.13 (d, 2H, *J_value_* = 15.2MHz, CH=CH), 7.50 (s, 1H, NH), 7.60–7.43 (m, 3H, Ar), 10.0 (s, 1H, OH). ^13^C NMR (100 MHz, DMSO-d_6_, δ): 165.8, 149.5, 147.9, 144.3, 138.1, 136.5, 129.4, 128.6, 126.1, 125.3, 124.7, 112.7, 111.9, 55.9, 22.7. Anal. Calcd. For C_17_H_17_NO_3_ (283.32): C, 72.07; H, 6.05; N, 4.94. Found: C, 72.37; H, 6.25; N, 4.74.

##### (E)-N-Cyclohexyl-3-(4-hydroxy-3-methoxyphenyl)acrylamide (15)

IR (KBr, cm^−1^): 3380, 3240 (OH and NH), 2920 (C-H aliphatic), 1655 (C=O). ^1^H NMR (DMSO-d_6_, D_2_O, δ): 1.05–1.71 (m, 10H, CH_2_), 3.37 (m, 1H, CH), 3.84 (s, 3H, OCH_3_), 5.70 and 5.72 (d, 2H, *J_value_* = 15.9 MHz, HC=CH), 6.76–7.26 (m, 3H, Ar-H + 1OH, 1NH, D_2_O exchangeable), ^13^C NMR (100 MHz, DMSO-d_6_, δ): 169.8, 148.9, 147.1, 143.3, 138.1, 129.4, 127.6, 125.1, 123.3, 122.7, 116.7, 110.9, 56.2, 50.7, 32.5, 24.9; Anal. Calcd. For C_16_H_21_NO_3,_ (275.34): C, 69.79; H. 7.69; N, 5.09. Found: C, 69.97; H. 7.79; N, 5.00.

##### (E)-3-(4-Hydroxy-3-methoxyphenyl)-N-(4-methoxyphenyl)acrylamide (16)

IR (KBr, cm^−1^): 3460, 3344 (OH and NH), 3050 (C-H aromatic), 2908, 2839 (C-H aliphatic), 1635 (C=O). ^1^H NMR (DMSO-d_6_, D_2_O, δ): 3.81 (s, 3H, OCH_3_), 3.84 (s, 3H, OCH_3_), 6.68 and 7.05 (d, 2H, *J_value_* = 15.4 MHz, CH=CH), 6.88–6.83 (m, 4H, Ar), 7.54–7.47 (m, 3H, Ar-H), 7.54 (s, 1H, NH), 10.01 (s, 1H, OH). ^13^C NMR (100 MHz, DMSO-d_6_, δ): 165.8, 148.5, 147.9, 144.3, 138.1, 130.3, 129.4, 127.6, 126.1, 124.3, 122.7, 114.7, 112.9, 56.2, 55.7; Anal. Calcd. For C_17_H_17_NO_4_ (299.32): C, 68.21; H, 5.72; N, 4.68. Found: C, 68.25; H, 5.63; N, 4.55.

##### (E)-3-(3-Hydroxy-4-methoxyphenyl)-N-(m-tolyl)acrylamide (17)

IR (KBr, cm^−1^): 3460, 3344 (OH and NH), 3050 (C-H aromatic), 2908 (C-H aliphatic), 1630 (C=O). ^1^H NMR (DMSO-d6, D_2_O, δ): 2.29 (s, 3H, CH_3_), 3.84 (s, 3H, OCH_3_), 6.68 and 7.06 (d, 2H, *J_value_* = 15.0 MHz, CH=CH), 6.88–6.83 (m,4H, Ar-H), 7.54–7.47 (m, 3H, Ar), 7.54 (s, 1H, NH), 10.01 (s, 1H, OH). ^13^C NMR (100 MHz, DMSO-d_6_, δ): 165.6, 149.3, 147.1, 143.3, 139.6, 128.4, 126.6, 126.1, 122.3, 121.7, 114.4, 113.7, 56.9; Anal. Calcd. For C_17_H_17_NO_3_ (283.32): C, 72.07; H, 6.05; N, 4.94. Found: C, 71.95; H, 6.32; N, 4.79.

##### (E)-N-Benzyl-3-(3-hydroxy-4-methoxyphenyl) acrylamide (18)

IR (KBr, cm^−1^): 3425, 3380 (OH and NH), 3120 (C-H aromatic), 2839 (C-H aliphatic), 1640 (C=O). ^1^H NMR (DMSO-d_6_, D_2_O, δ): 3.19 (s, 2H, CH_2_), 3.81 (s, 3H, OCH_3_), 6.57 and 6.83 (d, 2H, *J_value_* = 15.2 MHz, CH=CH), 7.15 (s, 1H, NH), 7.74–7.23 (m, 8H, Ar-H), 9.86 (s, H, OH). ^13^C NMR (100 MHz, DMSO-d_6_, δ): 167.6, 149.1, 147.9, 141.3, 137.6, 128.4, 126.6, 126.1, 122.3, 120.7, 114.4, 112.7, 56.4, 43.6; Anal. Calcd. For C_17_H_17_NO_3_ (283.32): C, 72.07; H, 6.05; N, 4.94. Found: C, 72.07; H, 6.05; N, 4.94.

##### (E)-N-(4-Chlorophenyl)-3-(4-hydroxy-3-methoxyphenyl)acrylamide (19)

IR (KBr, cm^−1^): 3460, 3290 (OH and NH), 3089 (C-H aromatic), 2846 (C-H aliphatic), 1630 (C=O). ^1^H NMR (DMSO-d_6_, D_2_O, δ): 3.83 (s, 3H, OCH_3_), 6.66–6.46 (m, 3H, Ar-H), 7.05 and 7.37 (d, 2H, *J_value_* = 14.7 MHz, CH=CH), 7.20 (s, 1H, NH), 7.15–7.40 (m, 4H, Ar), 9.55 (s, 1H, OH). ^13^C NMR (100 MHz, DMSO-d_6_, δ): 165.6, 149.1, 147.9, 141.3, 135.6, 133.5, 131.6, 129.7, 122.4, 121.9, 118.3, 116.7, 111.3, 56.1; Anal. Calcd. For C_16_H_14_ClNO_3_ (303.74): C, 63.27; H, 4.65; N, 4.61. Found: C, 63.27; H, 4.65; N, 4.61.

##### (E)-N-(2-Aminophenyl)-3-(3-hydroxy-4-methoxyphenyl)acrylamide (20)

IR (KBr, cm^−1^): 3447, 3345 (OH, NH and NH_2_), 3022 (C-H aromatic), 2974 (C-H aliphatic), 1642 C=O. ^1^H NMR (DMSO-d_6_, D_2_O, δ): 3.83 (s, 3H, OCH_3_), 4.94 (s, 1H, NH_2_), 6.61–6.37 (m, 4H, Ar-H), 6.75 and 6.85 (d, 2H, *J_value_* = 14.9 MHz, CH=CH), 7.20 (s, 1H, NH), 7.50–7.35 (m, 3H, Ar-H), 9.28 (s, 1H, OH), ^13^C NMR (100 MHz, DMSO-d_6_, δ):168.00, 149.1, 147.9, 147.2, 141.3, 127.7, 125.9, 125.3, 122.9, 118.3, 114.7, 112.3, 56.8; Anal. Calcd. For C_16_H_16_N_2_O_3_ (284.30): C, 67.59; H, 5.67; N, 9.85. Found: C, 67.77; H, 5.51; N, 9.73.

### 3.4. Biological Activity

#### 3.4.1. Cell Culture

MCF-7 cells (Human breast adenocarcinoma), HepG2 (human hepatocellular carcinoma), A549 (human lung adenocarcinoma), Caco-2 (colon carcinoma), PANC-1 (human pancreatic cancer), and Vero cells (derived from normal kidney cells) were supplied by (Holding Company for Biological Products and Vaccines VACSERA; Giza, Egypt). The cells were cultured in PRMI 1640 medium (Lonza, Switzerland) supplemented with 10% fetal bovine serum (Gibco), 1% penicillin and 1% streptomycin (Sigma Aldrich, Saint Louis, MO, USA) at 37 °C under an atmosphere of 5% CO_2_ and 95% air.

#### 3.4.2. Cell Viability Determination by MTT Assay

The effect of parent compounds and their derivatives were assessed by (3-(4,5-dimethylthiazol-2-yl)-2,5diphenyl-tetrazolium bromide) MTT assay. On a 96-well tissue culture plate, all cell lines were inoculated at a density of 1 × 10^5^ cells/mL (100 µL/well) and incubated at 37 °C for 24 h in order to form a complete monolayer sheet. After a merged sheet of cells developed, the growth media was decanted from the 96-well microtiter and refilled with different concentrations of compounds (µg/mL), and the plate was incubated at 37 °C for 48 h to minimize exposure to the dissolving agent [56] (Appendix A).

MTT solution (BIO BASIC CANADA INC) was prepared [5 mg/mL in phosphate buffered saline (PBS)], added to each well and incubated for 4 h, then after removing the media, 200 μL of DMSO was added to each well to solubilize the formazan crystals and the optical density of the formazan product was measured at 560 nm using microplate reader (mindray, MR-96A). The results were calculated as the mean of the three independent experiments.

#### 3.4.3. Annexin V/Propidium Iodide (PI) Apoptosis Assay by Flow Cytometry

In order to determine the percentage of viable, apoptotic and necrotic cells after treatment with our compounds, double-staining with fluorescein isothiocyanate (FITC) and PI was performed. Annexin V-FITC Apoptosis Detection Kit (BioVision, CA, USA) was used. Briefly, cultured Caco-2 were treated with the tested compounds or DMSO for 24h. Cells were then collected by centrifugation at 2000 rpm for 5 min, and the pellet was washed with PBS then centrifuged again at 2000 rpm for 5 min then resuspended in 500μL of the kit’s binding buffer before adding the Annexin V-FITC and PI. The cells were then incubated at room temperature in the dark for 5 min. Annexin V-FITC binding was measured using FACS Calibur flow cytometer (BD biosciences, USA) (Ex = 488 nm, Em = 533 nm, FL1 filter for annexin and FL2 filter for PI).

#### 3.4.4. Quantitative Polymerase Chain Reaction (qPCR) and RNA Extraction

The treated cells were harvested and the total RNA was extracted using RNeasy Mini Kit (Qiagen) according to the manufacturer’s instruction. The primers of the required genes (Table 6) were used for SYBR Green qRT-PCR using a one-step RT-PCR kit with SYBR Green (Bio-Rad). qRT-PCR was performed on Rotor Gene Q (Qiagen). The mRNA levels of p53, Bcl-2, Caspase-3, Bax, MMP-2, and MMP-9 were calculated using the 2^−ΔΔCT^ method with the endogenous β-actin mRNA as control to normalize the values. Rotor-Gene 6000 Series Software was used for data analysis. and their HGNC ID was added (Table 6)

## 4. Conclusions

Despite the high efficacy of CA and FA, they are limited by their high metabolism and consequent low bioavailability. Therefore, we were challenged to synthesize highly stable and biologically active compounds. Henceforth, we designed and investigated 18 congeners of amide nature; they were all superior to their parent acids in terms of cytotoxic activity against Caco-2. The selected candidates for cell-cycle analysis and gene expression assays were those displaying superior cytotoxicity and high SI using Vero normal cells.

Subsequently, the assays offered six remarkably active compounds, **7**, **10**, **11**, **14**, **15** and **17**, and five of them comprised an aromatic ring adjacent to the amide bridge. In accordance with the SAR, the highest cytotoxicity was evident amongst the aromatic derivative with *p*-chloro phenyl **10,** displaying excellent cytotoxic activity, this was deduced when the cytotoxicity decreased after the chloro residue was replaced with a bromo substitution. It could also be concluded that overall, the caffeic acid amides surpassed the ferulic analogues.

We are planning to further study other amide derivatives and assess their activity in a suitable animal model.

## Figures and Tables

**Figure 1 molecules-27-06724-f001:**
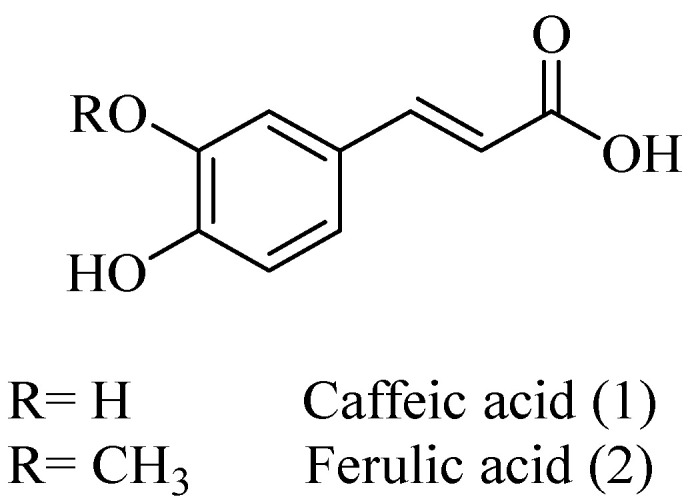
Structure of Caffeic acid and Ferulic acid.

**Figure 2 molecules-27-06724-f002:**
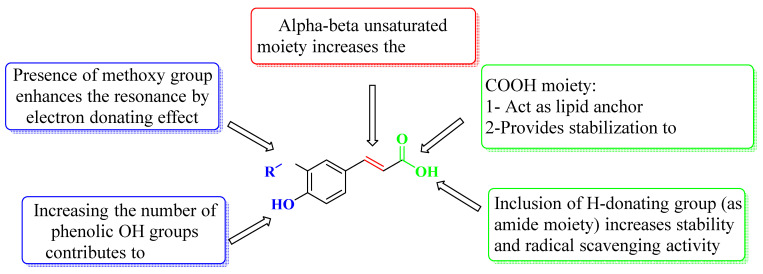
SAR of caffeic acid and ferulic acid as antioxidants.

**Figure 3 molecules-27-06724-f003:**
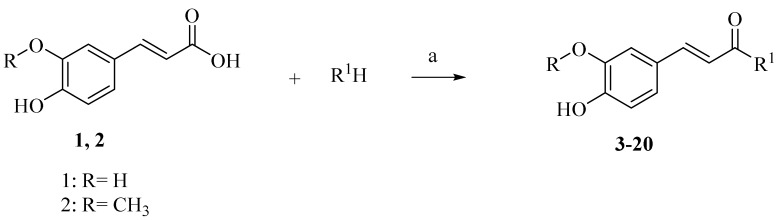
Scheme for the preparation of the candidate compounds.

**Figure 4 molecules-27-06724-f004:**
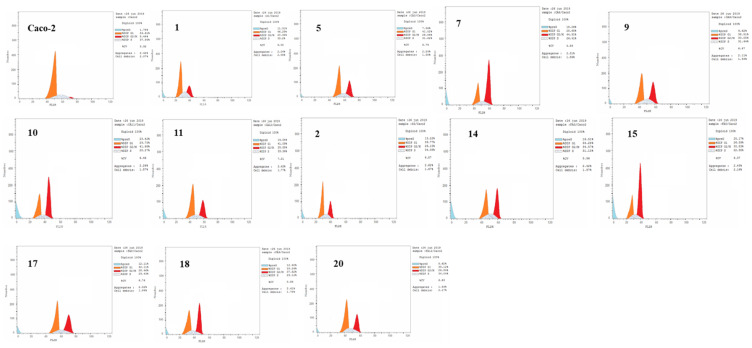
Cell-cycle analysis of Caco-2 cells following treatment with different derivatives and their parents; Histogram showing the percentages of cells in different cell cycle phases.

**Figure 5 molecules-27-06724-f005:**
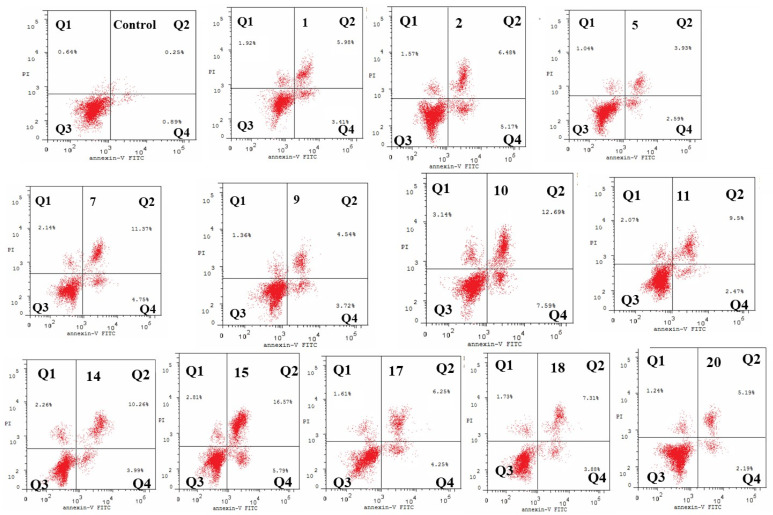
Dot plot representing four quadrant images observed by flow cytometry analysis. Q1: shows necrotic cells, Q2: shows later period apoptotic cells, Q3: shows normal cells and the Q4: shows early apoptotic cells.

**Figure 6 molecules-27-06724-f006:**
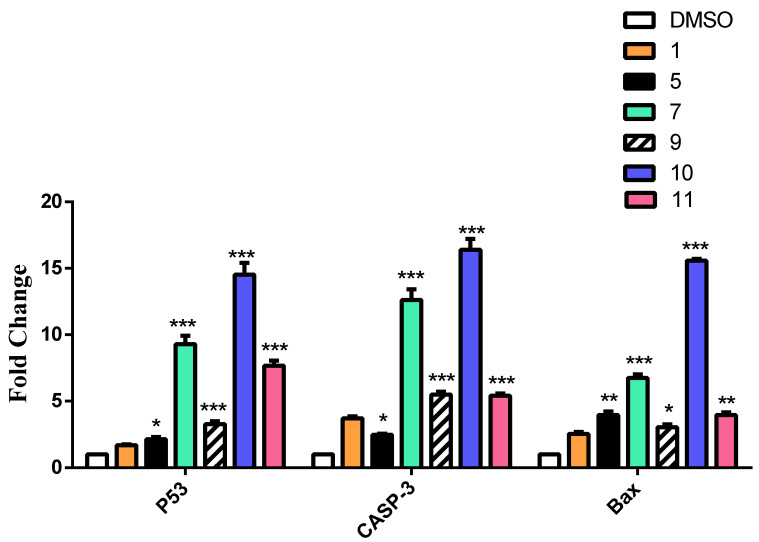
The effect of CA and its derivatives on the expression of P53, Caspase 3 and Bax genes by RT-qPCR. The results were represented as mean of fold changes ± SD. β-actin gene was used as a control gene. * Significant from CA at *p* < 0.05, ** Significant from CA at *p* < 0.01, *** Significant from CA at *p* < 0.001.

**Figure 7 molecules-27-06724-f007:**
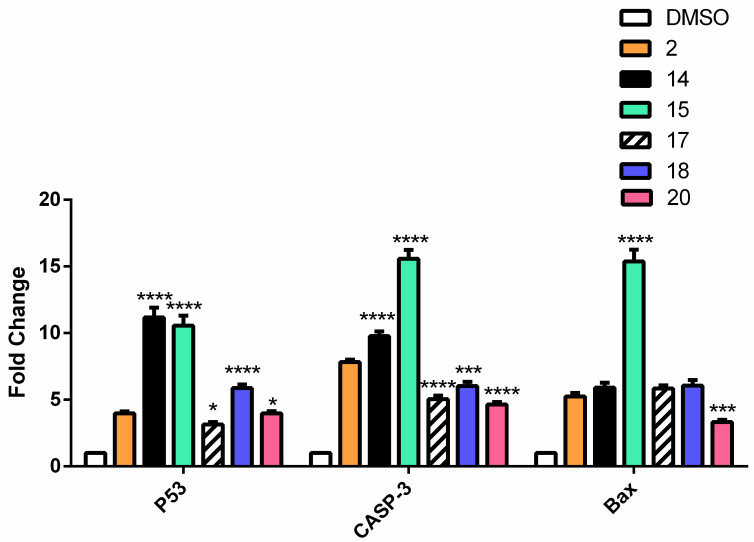
The effect of FA and its derivatives on the expression of P53, Caspase 3 and Bax genes by RT-qPCR. The results were represented as mean of fold changes ± SD. β-actin gene was used as a control gene. * Significant from FA at *p* < 0.05, *** Significant from FA at *p* < 0.001, ******** Significant from FA at *p* < 0.0001.

**Figure 8 molecules-27-06724-f008:**
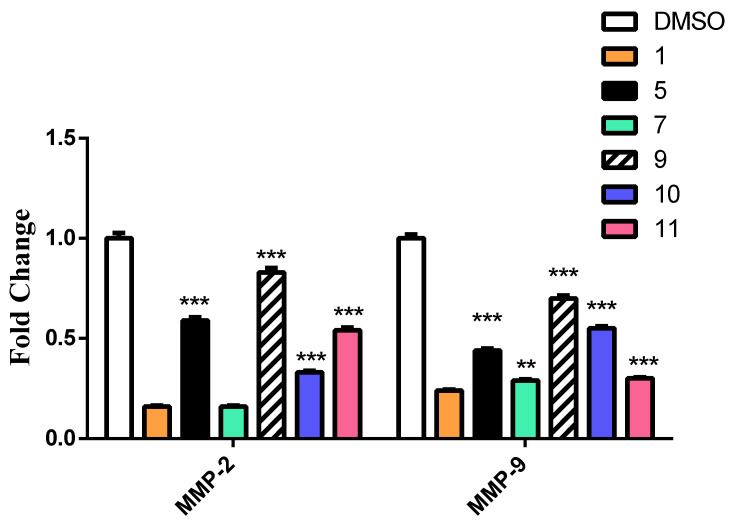
The effect of CA and its derivatives on the expression of MMP-2 and MMP-9 by RT-qPCR. The results were represented as mean of fold changes ± SD. β-actin gene was used as a control gene. ** Significant from CA at *p* < 0.01, *** Significant from CA at *p* < 0.001.

**Figure 9 molecules-27-06724-f009:**
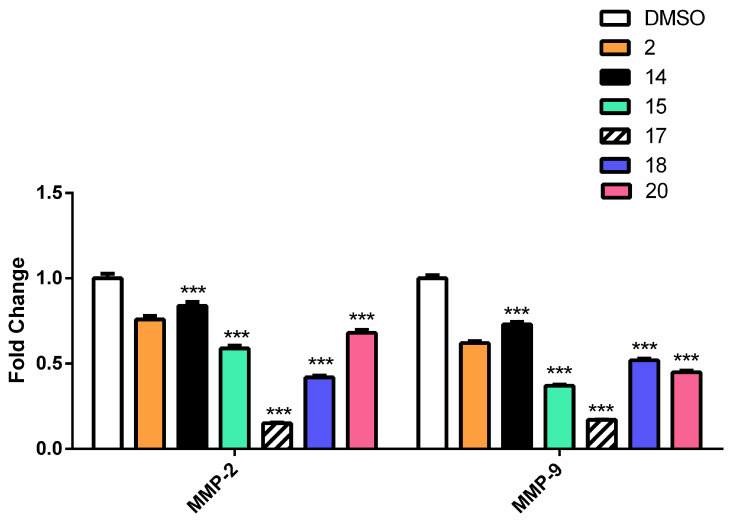
The effect of FA and its derivatives on the expression of MMP-2 and MMP-9 genes by RT-qPCR. The results were represented as mean of fold changes ± SD. β-actin gene was used as a control gene. *** Significant from FA at *p* < 0.001.

**Figure 10 molecules-27-06724-f010:**
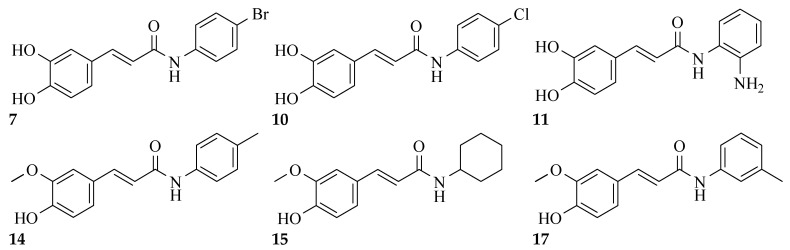
Structures of the most active compounds throughout all the biological tests.

**Table 1 molecules-27-06724-t001:** Detailed structure of the semi synthesized compounds **3**–**20**.

Cpd. No.	R	R^1^	Cpd. No.	R	R^1^
3	H	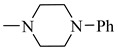	12	CH_3_	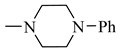
4	H	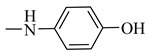	13	CH_3_	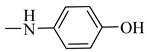
5	H	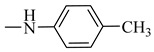	14	CH_3_	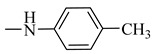
6	H	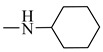	15	CH_3_	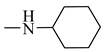
7	H	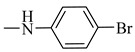	16	CH_3_	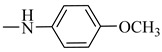
8	H	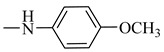	17	CH_3_	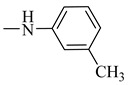
9	H	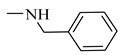	18	CH_3_	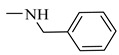
10	H	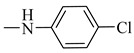	19	CH_3_	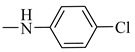
11	H	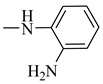	20	CH_3_	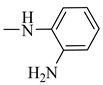

**Table 2 molecules-27-06724-t002:** IC_50_ of the parent compounds and doxorubicin on different cancer cell lines. Results are the average of triplicates.

Cpd. No.	IC_50_ (μM) ± SD					SI *
	MCF-7	HepG2	A549	Caco-2	PANC-1	Vero
**Caffeic acid (1)**	0.97 ± 6.27	1.03 ± 12.63	1.63 ± 15.56	0.41 ± 3.3	1.29 ± 12.16	0.90
**Ferulic acid (2)**	1.12 ± 10.15	0.81 ± 2.17	1.74 ± 10.8	0.41 ± 1.92	1.08 ± 2.27	1.08
**Doxorubicin**	0.17 ± 2.00	0.18 ± 3.4	0.14 ± 1.3	0.17 ± 1.1	0.16 ± 1.86	0.04

* SI (Selectivity index) = IC_50_ on Vero cells/IC_50_ on cancer cells.

**Table 3 molecules-27-06724-t003:** IC_50_ of the semi-synthesized compounds on Caco-2 cancer cell line and the selectivity index on Vero normal cell line. Results are the average of triplicates.

Cpd. No.	IC_50_ (μM)	SI Vero	Cpd. No.	IC_50_ (μM)	SI * Vero
**3**	0.08 ± 0.62	0.087	**12**	0.04 ± 0.34	0.048
**4**	0.07 ± 0.36	0.094	**13**	0.08 ± 1.2	0.089
**5**	0.06 ± 0.35	0.34	**14**	0.05 ± 0.59	0.18
**6**	0.05 ± 0.44	0.083	**15**	0.02 ± 0.17	0.13
**7**	0.02 ± 0.11	0.99	**16**	0.08 ± 0.69	0.13
**8**	0.08 ± 0.32	0.14	**17**	0.02 ± 0.21	0.12
**9**	0.03 ± 0.29	0.10	**18**	0.05 ± 0.86	0.15
**10**	0.009 ± 0.09	0.13	**19**	0.02 ± 0.65	0.077
**11**	0.01 ± 0.11	0.13	**20**	0.01 ± 0.63	0.082

* SI (Selectivity index) = IC_50_ on Vero cells/IC_50_ on Caco-2.

**Table 4 molecules-27-06724-t004:** Cell-cycle analysis of Caco-2 cells following treatment with the parent compounds and their derivatives.

Sample Data	Results			
Cpd. No.	%G0–G1	%S	%G2/M	%Pre-G1
**Control (Caco-2)**	53.61 ± 3.16	37.95 ± 2.25	8.44 ± 0.63	1.78 ± 0.08
**1**	46.29 ± 3.1	33.22 ± 1.99	20.49 ± 1.92	11.31 ± 0.81
**2**	39.77 ± 2.29	34.08 ± 2.63	26.15 ± 3.26	13.22 ± 0.63
**5**	41.52 ± 2.61	31.62 ± 1.79	26.86 ± 2.75	7.56 ± 0.39
**7**	28.68 ± 1.64	26.41 ± 2.15	44.91 ± 3.42	18.26 ± 1.42
**9**	38.51 ± 1.95	31.44 ± 2.87	30.05 ± 2.59	9.62 ± 0.37
**10**	29.75 ± 1.73	28.27 ± 1.99	41.98 ± 2.18	23.42 ± 2.12
**11**	41.09 ± 2.76	33.36 ± 2.73	25.55 ± 2.47	14.04 ± 0.95
**14**	34.28 ± 1.74	31.15 ± 3.17	34.57 ± 4.16	16.51 ± 0.63
**15**	26.59 ± 2.67	22.58 ± 2.91	50.83 ± 3.49	25.17 ± 3.29
**17**	42.11 ± 3.41	29.43 ± 3.16	28.46 ± 2.64	12.11 ± 0.97
**18**	33.26 ± 4.29	29.12 ± 4.05	37.62 ± 2.47	12.92 ± 2.16
**20**	39.12 ± 2.58	34.04 ± 2.69	26.84 ± 2.34	8.62 ± 0.69

**Table 5 molecules-27-06724-t005:** Detection of different types of apoptotic cells induced in Caco-2 cells following treatment with parent compounds and their derivatives using annexin-V FITC/PI staining.

Cpd. No.	Total	Early Apoptosis	Late Apoptosis	Necrosis
**Control (Caco-2)**	**1.78**	**0.89**	**0.25**	**0.64**
**1**	11.31	3.41	5.98	1.92
**2**	13.22	5.17	6.48	1.57
**5**	7.56	2.59	3.93	1.04
**7**	18.26	4.75	11.37	2.14
**9**	9.62	3.72	4.54	1.36
**10**	23.42	7.59	12.69	3.14
**11**	14.04	2.47	9.50	2.07
**14**	16.51	3.99	10.26	2.26
**15**	25.17	5.79	16.57	2.81
**17**	12.11	4.25	6.25	1.61
**18**	12.92	3.88	7.31	1.73
**20**	8.62	2.19	5.19	1.24

**Table 6 molecules-27-06724-t006:** Primers used in qRT-PCR.

Gene	Primers Used
**p53**	F 5′-CCCCTCCTGGCCCCTGTCATCTTC-3′ R 5′-GCAGCGCCTCACAACCTCCGTCAT-3′
**Bcl-2**	F 5′-CCTGTGGATGACTGAGTACC-3′ R 5′-GAGACAGCCAGGAGAAATCA-3′
**Caspase 3**	F 5′-TTCATTATTCAGGCCTGCCGAGG-3′ R 5′-TTCTGACAGGCCATGTCATCCTCA-3′
**Bax**	F 5′-GTTTCATCCAGGATCGAGCAG-3′ R 5′-CATCTTCTTCCAGATGGTGA-3′
**MMP-2**	F 5′- CAAAAACAAGAAGACATACATCTT-3′ R 5′- CAAAAACAAGAAGACATACATCTT-3′
**MMP-9**	F 5′- TGGGGGGCAACTCGGC-3′ R 5′- GGAATGATCTAAGCCCAG-3′
**β-actin**	F 5′-GTGACATCCACACCCAGAGG-3′ R 5′-ACAGGATGTCAAAACTGCCC-3′

## Data Availability

Not applicable.

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
