# Peer review of "Development of Semisynthetic Apoptosis-Inducing Agents Based on Natural Phenolic Acids Scaffold: Design, Synthesis and In-Vitro Biological Evaluation"

_molecules, 2022, doi:10.3390/molecules27196724_

Round 1

Reviewer 1 Report (Previous Reviewer 2)

The manuscript entitled "Development of Semisynthetic Apoptosis-Inducing Agents Based on Natural Phenolic Acids Scaffold: Design, Synthesis and in-vitro Biological Evaluation” evaluates the biological anticancer activity of some semisynthetic derivatives of caffeic and ferulic acids. Biological studies included basic viability evaluation screening in five human cancer cell lines as well as some preliminary mechanism of action evaluation in vitro, including the cell cycle profile, the gene expression of some apoptosis and metastasis-enhancing genes. The synthetic methodology is well-characterized. The biological experiments performed are sufficient for preliminary compound evaluation. The conclusions are appropriate and are based on the current study results.

The manuscript contains a set of valuable data, both from a chemical and biological point of view, however, some improvements are necessary for the manuscript. 

Comments:     

1)     In section 3.2.3. (Gene Expression), have the authors considered that the gene expression assays using RT-PCR may not be adequate for quantifying the target apoptosis-related signals since the mRNA expression may not necessarily reflect the corresponding protein levels due to the post-translational modifications. In fact, detecting protein signals using ELISA or Western blotting or enzymatic activity of caspase-3, for example, is expected to give more reliable data than the gene expression assays.

The authors are advised to add the response to this point within the context of the discussion section.

2) Again, in section 3.2.3.2. (Caffeic acid and ferulic acid derivatives inhibit metastasis enhancing genes), detecting MMP-2 and MMP-9 at the gene expression level may not be adequate. Instead, detecting the activity of MMP-2 and MMP-9 is more predictive regarding the potential metastasis.

The authors are advised to add the response to this point within the context of the discussion section.

3) In section 2.4.1. (Cell viability determination by MTT assay). Why was incubation time (48 h) used in the manuscript? Treatment for only 48 h seems unable to give an enough picture of the anti-cancer effect of candidate drugs. In general, a time course covering up to 24 or 72 h should be expected.

The authors are advised to add the response to this point within the context of the material and methods section.

4) In figures 5-8, the “control cell” should be replaced by the name of the vehicle used for solubilization of tested agents since all compounds were tested on the same cell line.

5) The authors have added several supplementary files which document the chemical identity of the investigated compounds alongside the biological reports for RT-PCR and flow cytometry assays. However, the supplementary files need to be carefully labeled, not just by adding files named 12, 13, 14, 15, …etc.    

Author Response

Comments of reviewer 1    

1)     In section 3.2.3. (Gene Expression), have the authors considered that the gene expression assays using RT-PCR may not be adequate for quantifying the target apoptosis-related signals since the mRNA expression may not necessarily reflect the corresponding protein levels due to the post-translational modifications. In fact, detecting protein signals using ELISA or Western blotting or enzymatic activity of caspase-3, for example, is expected to give more reliable data than the gene expression assays.

The authors are advised to add the response to this point within the context of the discussion section.

 Done

2) Again, in section 3.2.3.2. (Caffeic acid and ferulic acid derivatives inhibit metastasis enhancing genes), detecting MMP-2 and MMP-9 at the gene expression level may not be adequate. Instead, detecting the activity of MMP-2 and MMP-9 is more predictive regarding the potential metastasis.

The authors are advised to add the response to this point within the context of the discussion section.

 Done

3) In section 2.4.1. (Cell viability determination by MTT assay). Why was incubation time (48 h) used in the manuscript? Treatment for only 48 h seems unable to give an enough picture of the anti-cancer effect of candidate drugs. In general, a time course covering up to 24 or 72 h should be expected.

The authors are advised to add the response to this point within the context of the material and methods section.

 Done

4) In figures 5-8, the “control cell” should be replaced by the name of the vehicle used for solubilization of tested agents since all compounds were tested on the same cell line.

 Done

5) The authors have added several supplementary files which document the chemical identity of the investigated compounds alongside the biological reports for RT-PCR and flow cytometry assays. However, the supplementary files need to be carefully labeled, not just by adding files named 12, 13, 14, 15, …etc.    

 Done

Reviewer 2 Report (Previous Reviewer 1)

 Shahira M. Ezzat reported that CA and FA are limited by their high metabolism 546 and consequent low bioavailability. Therefore, they synthesize highly 547 stable and biologically active compounds. The authors improved the paper. I think that would be interesting to show, if possible p53 or caspase3 also by western blotting and not only by PCR.

Author Response

 Regarding the above points we totally agree with the reviewer. The point is, we aimed in this paper to screen for the compounds that possess potential cytotoxic activity for the use as anticarcinogenic molecules and the most sensitive cell line that could be used for our testing. We also aimed at ruling out the possibility of cell death via necrosis. Phase 2 of the project will involve most of the proposed experiments, where caspases 3,7 and 9 will be assayed via western blotting, along with assaying mitochondrial damage via cytochrome c. In addition, we will aim at testing the compound vs normal human cell lines

This manuscript is a resubmission of an earlier submission. The following is a list of the peer review reports and author responses from that submission.

Round 1

Reviewer 1 Report

The paper of Ezzat et al., compared caffeic 23 acid (CA), and ferulic acid (FA) derivates  to improve these compounds pharmacological activities. d very potent 27. Eighteen derivatives were tested for their anticancer activity against Caco-2 cell line, which was the most responsive compared to others 5 cell culture lines.

Major revision:

1.       The introduction is too long, should be revised

2.        MTT assay is not sufficient, should be added another cell viability test.

3.       Apoptosis was demonstrated only with flow cytometry. Authors should add TUNEL assay and western blotting for caspase. The apoptosis is mediated by the intrinsic or extrinsic pathway?

4.       The induction in casp3 expression, doesn’t mean an increase in caspase 3 clivage/activation, any data about the contribution of more caspase 3 protein level and the increase in cleavage and apoptosis?

5.       You should do some siRNA experiment to figure out if BAX or P53 induction are responsible of apoptosis

Reviewer 2 Report

The manuscript entitled "Development of Semisynthetic Apoptosis Inducer Agents Based on Natural Phenolic Acids Scaffold: Design, Synthesis and in-vitro Biological Evaluation” evaluates the biological anticancer activity of some semisynthetic derivatives of caffeic and ferulic acids. Biological studies included basic viability evaluation screening in five human cancer cell lines as well as some preliminary mechanism of action evaluation in vitro, including the cell cycle profile, the gene expression of some apoptosis and metastasis enhancing genes. The synthetic methodology is well-characterized. The biological experiments performed are sufficient for preliminary compound evaluation. The conclusions are appropriate and are based on study results.

Comments:     

1)      In section 2.2.3. (Gene Expression), have the authors considered that the gene expression assays using RT-PCR may not be adequate for quantifying the target apoptosis-related signals since the mRNA expression may not necessarily reflect the corresponding protein levels due to the post-translational modifications. In fact, detecting protein signals using ELISA or Western blotting or enzymatic activity of caspase-3, for example, is expected to give more reliable data than the gene expression assays. This point needs to be addressed in the discussion section.

2)      Again, in section 2.2.3.2. (Caffeic acid and ferulic acid derivatives inhibit metastasis enhancing genes), detecting MMP-2 and MMP-9 at the gene expression level may not be adequate. Instead, detecting the activity of MMP-2 and MMP-9 is more predictive regarding the potential metastasis. This point needs to be addressed in the discussion section.

3)      In section 4.4.1. (4.4.1. Cell viability determination by MTT assay). Why was incubation time (48 h) used in the manuscript? Treatment for only 48 h seems not able to give an enough picture of the anti-cancer effect of candidate drugs. In general, a time course covering up to 24 or 72 h should be expected.

4)      In section 2.2.1 (Cell viability determination by MTT assay), the authors are advised to describe the rationale for using doxorubicin as the reference anticancer agent.

5)      In section 2.2.2.1 (Flow cytometry analysis showing induction of apoptosis). Authors are advised to clarify for readers that the increase in % of cell population at the Pre-G1 phase is indicative of apoptosis. This needs to be incorporated in line 203.

6)      The resolution for the pictures displayed in figure 3 and figure 4 is very low and is not clear to readers. Authors are advised to upload magnified and high-resolution pictures for figures 3 and 4.

7)      In figures 5-8, the “control cell” should be replaced by the name of the vehicle used for solubilization of tested agents since all compounds were tested on the same cell line.

 8)    The title "Development of Semisynthetic Apoptosis Inducer Agents Based on Natural Phenolic Acids Scaffold: Design, Synthesis and in-vitro Biological Evaluation” needs to be corrected where “apoptosis induced agents” should be replaced by “apoptosis-inducing agents”.

 9)    In figures 5-8, the authors are advised to unify the use of colored versions of these figures.  

Round 2

Reviewer 1 Report

I disagree with the authors' statement "This work aimed to test biological activity and not the mechanism of derivatives". The authors are starting to show mechanistic data, and I think adding some exponents could improve the paper because in the latter version, a central part of the paper is missing. The fact that no experiments were done and the points requested were hastily answered togheter shows little desire to improve the work and for me it is not allowed. It was enough that at least some experiments of those requested had been made. The answer shows little willingness to do further work and it is very disappointing.

Author Response

Please see the attchment
